# The Impact of Antiosteoporotic Drugs on Glucose Metabolism and Fracture Risk in Diabetes: Good or Bad News?

**DOI:** 10.3390/jcm10050996

**Published:** 2021-03-02

**Authors:** Athanasios D. Anastasilakis, Elena Tsourdi, Gaia Tabacco, Anda Mihaela Naciu, Nicola Napoli, Fabio Vescini, Andrea Palermo

**Affiliations:** 1Department of Endocrinology, 424 General Military Hospital, 56429 Thessaloniki, Greece; a.anastasilakis@gmail.com; 2Department of Medicine (III) &Center for Healthy Aging, Technische Universität Dresden Medical Center, 01307 Dresden, Germany; 3Unit of Endocrinology and Diabetes, Campus Bio-Medico University, 00128 Rome, Italy; g.tabacco@unicampus.it (G.T.); a.naciu@unicampus.it (A.M.N.); N.Napoli@unicampus.it (N.N.); A.Palermo@unicampus.it (A.P.); 4Department of Endocrinology and Diabetes, Santa Maria della Misericordia Hospital, 33100 Udine, Italy; fvescini@alice.it

**Keywords:** diabetes mellitus, fracture, bisphosphonates, denosumab, teriparatide, osteoporosis, bone turnover

## Abstract

Osteoporosis and diabetes mellitus represent global health problems due to their high, and increasing with aging, prevalence in the general population. Osteoporosis can be successfully treated with both antiresorptive and anabolic drugs. While these drugs are clearly effective in reducing the risk of fracture in patients with postmenopausal and male osteoporosis, it is still unclear whether they may have the same efficacy in patients with diabetic osteopathy. Furthermore, as bone-derived cytokines (osteokines) are able to influence glucose metabolism, it is conceivable that antiosteoporotic drugs may have an effect on glycemic control through their modulation of bone turnover that affects the osteokines’ release. These aspects are addressed in this narrative review by means of an unrestricted computerized literature search in the PubMed database. Our findings indicate a balance between good and bad news. Active bone therapies and their modulation of bone turnover do not appear to play a clinically significant role in glucose metabolism in humans. Moreover, there are insufficient data to clarify whether there are any differences in the efficacy of antiosteoporotic drugs on fracture incidence between diabetic and nondiabetic patients with osteoporosis. Although more studies are required for stronger recommendations to be issued, bisphosphonates appear to be the first-line drug for treatment of osteoporosis in diabetic patients, while denosumab seems preferable for older patients, particularly for those with impaired renal function, and osteoanabolic agents should be reserved for patients with more severe forms of osteoporosis.

## 1. Introduction


***“La sapienza è figliuola della sperienza”***


This quote by Leonardo da Vinci means “wisdom is the daughter of experience,” and nothing is truer when approaching the complex interplay between bone and glucose metabolism.

Osteoporosis and diabetes mellitus (DM) represent global health problems due to their high prevalence in the general population, and significant effort has been made in order to effectively manage these diseases. Osteoporosis can be successfully treated with both antiresorptive and anabolic drugs, whose efficacy in reducing fracture incidence has been well documented in several large, randomized control trials (RCTs). All antiosteoporotic drugs are also able to increase bone mineral density (BMD), although with some variation in the magnitude of this effect, and BMD represents a surrogate endpoint for RCTs on osteoporosis. In clinical practice, an increase in BMD is a sign of an adequate response to therapy, and it has been demonstrated that by increasing BMD, a significant reduction in fracture risk can be achieved [1].

Diabetic osteopathy is mainly characterized by deterioration of bone microstructure and of the so-called bone quality. Both type 1 (T1DM) and type 2 diabetes mellitus (T2DM) have been associated with impaired bone quality and increased fracture risk. Patients with T1DM have an overall reduced BMD and multifold increased risk for fractures compared with individuals without diabetes. Low bone turnover with reduced bone formation and, to a lesser degree, bone resorption, low levels of insulin and insulin-like growth factor-1 (IGF1), and accumulation of advanced glycation end products (AGEs) in collagen as a result of hyperglycemia are considered the main mechanisms of osteopathy in this context. In contrast to T1DM, in patients with T2DM, BMD is frequently normal, or even elevated, however, fracture risk is also increased [2]. Therefore, when considering the fracture risk of T2DM patients, BMD measurement is likely to be misleading. Given the important pathogenetic differences between diabetic osteopathy and postmenopausal osteoporosis, it is possible that antiosteoporotic drugs may have different efficacy in these two forms of bone disease.

Several preclinical studies on cell or animal models have clearly shown that some bone-derived factors, called osteokines, may influence glycemic control. The bone remodeling process is characterized by a continuous release of osteokines into the bloodstream. Therefore, by inhibiting or stimulating bone turnover with antiosteoporotic drugs, one may expect possible effects not only on BMD and fractures, but also on glucose homeostasis.

The effects of antiosteoporotic drugs on glucose metabolism, as well as their efficacy to reduce fracture risk in diabetic patients with osteoporosis, are addressed in this narrative review. It stands to reason that both answers should be affirmative, but as da Vinci taught us, we need evidence before drawing a conclusion.

## 2. Literature Search

Although this is not a systematic review, an unrestricted computerized literature search was performed in the PubMed database up to the 30 November 2020 to investigate the following topics: (1) the impact of antiosteoporotic drugs on glucose metabolism; (2) the impact of antiosteoporotic drugs on fracture risk in patients with DM. Based on Medical Subject Heading (MeSH) terms, the following query was set: “(“Osteoporosis, Postmenopausal/therapy”[Mesh]) AND “Diabetes Mellitus”[Mesh]” OR “glucose” “[Mesh].” We searched for articles published in English, and papers were excluded if they met the following criteria: review articles, case reports or series, and preclinical studies on cell or animal models.

Subsequently, an automatic alert was activated in PubMed (“My NCBI”) to retrieve relevant articles published after the initial search. Two investigators (G.T. and A.M.N.) independently searched for papers, screened the titles and abstracts of the retrieved articles, reviewed the full texts, and selected articles for inclusion. In case of disagreement, definitive reporting was achieved by mutual consensus.

## 3. How Bone Active Therapies Affect Glucose Metabolism

Osteoclast-mediated bone resorption and osteoblast-driven bone formation are pivotal in the maintenance and balance of bone remodeling. In particular, the skeleton comprises bone multicellular units (BMUs). A team of osteoclasts resorb a volume of old bone and, subsequently, osteoblasts deposit an equal volume of new bone at the same location to complete a cycle of remodeling. Bone remodeling is orchestrated by the osteocytes, which are osteoblasts that become entombed during the process of bone deposition. It has been well documented that osteocytes coordinate the function of osteoblasts and osteoclasts in response to both mechanical and hormonal stimuli by releasing factors such as sclerostin [3]. An imbalance between resorption and formation, in favor of the former, is the main mechanism leading to osteoporosis.

Antiresorptive drugs such as bisphosphonates (Bps) and denosumab (Dmab) reduce fracture risk by lowering the rate of bone remodeling until fewer BMUs are available to remove bone. Conversely, anabolic agents such as teriparatide (PTH 1–34) are able to reduce fracture risk by stimulating new bone formation in an attempt to directly restore bone volume and microstructure.

Over the last 15 years, an interplay between glucose metabolism and bone remodeling, mainly by the direct action of osteokines on the metabolic and glycemic pathways (Figure 1), has been proposed [4].

### 3.1. Osteokines and Glucose Homeostasis

The uncarboxylated form of the osteoblast-secreted molecule **osteocalcin** is a hormone that is able to partially regulate β-cell and adipocyte gene expression, leading to an improvement in glucose tolerance and an increase in energy expenditure [5]. Ferron et al. provided evidence that daily injections of uncarboxylated osteocalcin can improve glucose handling and prevent the development of T2DM in mice consuming a high-fat diet [6]. Clinical cross-sectional studies have also confirmed the association between osteocalcin and glucose tolerance and fat mass [7]. However, contrasting results regarding the ability of osteocalcin to improve glucose control and to decrease the risk of DM have been reported in meta-analyses [8,9]. Of note, many of the cross-sectional studies investigating osteocalcin and glycemic markers included both a nondiabetic and a diabetic group and investigated associations across the entire cohort. As in T2DM bone turnover marker levels are decreased compared to nondiabetic subjects, it cannot be excluded that associations between osteocalcin and glycemic markers could be driven by the presence of diabetes or not in such studies. Low osteocalcin levels are linked with impaired glucose metabolism in men and premenopausal women [10]. In particular, serum osteocalcin concentrations seem to have a negative and independent relation with HbA1c levels in men and women aged < 50 years, but not in postmenopausal women [11,12,13]. Moreover, the difference in osteocalcin concentrations between osteoporotic and nonosteoporotic males remains unclear but recent meta-analysis showed no significant difference [14]. The same results have been found between osteoporotic and nonosteoporotic females [15]. However, some authors showed that in elderly men and postmenopausal women with osteoporosis, higher osteocalcin concentrations were associated with lower BMD [16]. Besides its effects on glucose and energy expenditure, osteocalcin has also been reported to affect exercise capacity, male fertility, brain development, and cognition [17]. The undercarboxylated osteocalcin appears to affect glucose metabolism and testosterone synthesis, confirming that bone and organs such as pancreas and testis are connected by the bone-derived hormone osteocalcin [18]. On pancreatic β cells it favors insulin secretion, on muscle and white adipose tissue it stimulates glucose homeostasis, and on Leydig cells of the testis it promotes testosterone biosynthesis in response to its connection to the GPRC6A receptor [19].

Another osteokine that plays an important role in glucose metabolism is the receptor activator of nuclear factor kappa-Β ligand (**RANKL**), a member of the tumor necrosis factor superfamily that binds to RANK on cells of the myeloid lineage and functions as a key factor in osteoclast differentiation and activation. RANKL can deteriorate muscle strength, worsen insulin sensitivity [20], and increase energy expenditure by inducing beige adipocyte differentiation of preadipocytes in mice models [21]. Conversely, **osteoprotegerin** (OPG), also known as osteoclastogenesis inhibitory factor, enhances β-cell proliferation (without promoting differentiation) with a consequent increase in beta-cell mass, resulting in significantly delayed hyperglycemia in diabetic mice [22]. In contrast, cross-sectional evaluations of patients with prediabetes or overt diabetes depicted significantly higher serum concentrations of OPG in patients with these pathological conditions compared to subjects with normal glucose tolerance [23].

Genome-wide scans of subjects with diabetes have uncovered several genes associated with susceptibility to T2DM, including genes of Wnt signaling. As the Wnt/β-catenin signaling pathway is one of the regulators of not only osteogenesis but also adipogenesis, energy expenditure, and insulin sensitivity [24], its powerful natural inhibitor **sclerostin (SOST)** could play an important role in the crosstalk between bone and glucose metabolism. In particular, SOST KO mice showed a reduced adipogenesis and increased insulin sensitivity while sclerostin overproduction results in the opposite metabolic phenotype with adipocyte hypertrophy [25]. Different in vivo human studies have investigated sclerostin concentrations in subjects with impaired glucose metabolism. Indeed, sclerostin has been reported to positively affect glucose metabolic control in patients with T1DM [26], and a negative association with insulin resistance and sclerostin levels in obese subjects has been found [27]. Moreover, T2DM subjects have higher sclerostin levels than patients with T1DM [28,29,30,31], although this does not seem to be correlated with the risk of T2DM onset [27]. Preclinical findings also suggest that high glucose levels can directly increase the sclerostin expression leading to a negative impact on bone quality in DM [32]. Recent clinical data have confirmed that subjects with T2DM have higher bone gene expression of SOST, suggesting that impaired WNT signaling may be one of the key factors impairing bone metabolism in diabetic patients [33].

Another endogenous inhibitor of Wnt signaling is **dickkopf-1** (DKK-1). High circulating levels of DKK-1 have been found in T1DM [34] and T2DM subjects with endothelial dysfunction/platelet activation and cardiovascular disease [35]. Furthermore, glycated hemoglobin (HbA1c) was found to be negatively associated with DKK-1 in patients with T1DM [34].

It has also been well documented that alterations in the genes encoding low-density lipoprotein-related receptors 5 and 6 (**LRP5/6**) or their interacting proteins, which are proteins involved in pivotal functions of the canonical Wnt signaling, are linked to human diseases such as diabetes mellitus [36]. Thus, there are ample data confirming the involvement of Wnt signaling in the modulation of glucose metabolism.

Another category of osteokines with receptors in multiple organs, that are therefore potentially able to influence several metabolic functions not localized at the skeleton, is the bone morphogenetic proteins (**BMPs**). This group of growth factors (members of the transforming growth factor (TGF)-β superfamily) is released from the bone matrix into the circulation during bone resorption and affects both osteoclasts and osteoblasts. BMPs have recently been implicated in pancreas development, control of adult glucose homeostasis, energy expenditure by stimulation of brown adipose tissue formation [37], and the development of diabetic complications [38]. In particular, BMP-9 [39] and BMP-6 [40] improve glycemia and insulin resistance in T2DM mice and regulate glucose metabolism in hepatocytes. An in vivo human study confirmed that circulating BMP-9 levels are significantly higher in healthy subjects than in patients with newly diagnosed T2DM, and found a negative correlation with metabolic control, as assessed by HbA1c and fasting blood glucose [41].

### 3.2. Antiosteoporotic Treatments and Glucose Homeostasis

Most of the afore-mentioned evidence regarding the interplay between osteokines and glucose metabolism is derived from preclinical studies, with fewer clinical studies available. Based on this experimental evidence, it would appear feasible that antiosteoporotic treatments could alter glucose homeostasis by modulating bone turnover and subsequently the expression or release of the above-mentioned osteokines. However, most of the published clinical studies did not detect a significant impact of bone active therapies on glucose metabolism.

Primary hyperparathyroidism, a condition of chronically elevated parathyroid hormone (PTH) levels in the presence of hypercalcemia, has been linked to adverse effects on glucose metabolism (25). Few clinical studies support a weak, subclinical effect of PTH analogs such as **teriparatide** (TPTD) on glucose metabolism. In particular, daily subcutaneous injections of 20 µg of TPTD in women with severe postmenopausal osteoporosis seem to induce an acute transient worsening of the response to oral glucose administration that subsides after chronic use of TPTD [42]. A neutral effect of exogenous intermittent recombinant human PTH 1–34 administration on glucose homeostasis has been also reported by Anastasilakis et al. [43]. In contrast, Celer et al. reported that intermittent TPTD treatment may adversely affect fasting plasma glucose and insulin resistance, even six months after the beginning of anabolic therapy [44]. Of note, TPTD has been shown to be associated with improved glucose metabolic control in subjects with glucocorticoid-induced osteoporosis compared to treatment with Bps [45]. Thus, clinical studies have yielded discrepant results with regard to the effect of TPTD on glucose metabolism. Moreover, the mechanisms that explain TPTD-induced effects on glucose metabolism have not been clearly elucidated. It has been hypothesized that TPTD-induced impaired insulin resistance may be due to an increased intracellular free calcium concentration, leading to a decrease in insulin-dependent glucose transport [46], downregulation of insulin receptors [47], and increased islet amyloid polypeptide (IAPP) levels [48].

Different findings have been reported with the use of the full-length molecule **PTH (1–84)**. In detail, treatment with PTH (1–84) increased the concentration of both total and uncarboxylated osteocalcin forms, and decreased blood glucose, without influencing insulin secretion or resistance and pancreatic β-cell function [49]. This effect on glucose metabolism seems to be partially explained by the effect of PTH (1–84) on bone turnover [49].

In addition, incongruent findings have been reported regarding the interactions between antiresorptive therapies and glucose metabolism. Although some retrospective, population-based studies have shown a decreased risk of diabetes with the use of Bps [50,51], registry studies of zoledronate (ZOL) and alendronate (ALN) have not confirmed a positive effect on diabetes incidence [52]. Similarly, once-weekly treatment with risedronate was not associated with changes in glucose metabolism [53]. However, small prospective studies have demonstrated that ALN may improve fasting plasma glucose, HbA1c, and insulin indices in prediabetic, osteopenic, postmenopausal women [54] and reduce daily needs for insulin in patients with T1DM and osteoporosis [55].

As RANKL can be responsible for a deterioration in insulin sensitivity [20], it has been proposed that **Dmab** (a RANKL inhibitor) may have a favorable effect on glucose homeostasis. However, the use of Dmab has not been associated with a significant beneficial effect on fasting glucose in postmenopausal, osteoporotic women with prediabetes or diabetes, as clearly shown in the post hoc analysis of FREEDOM [52,56]. Other small prospective evaluations in nondiabetic, postmenopausal women with osteoporosis treated with Dmab have not shown any significant changes in glucose metabolism parameters [57,58]. Recently, it was demonstrated that Dmab may induce a short-term positive effect on insulin resistance in postmenopausal women [20] and women with breast cancer treated with an aromatase inhibitor [59]. In agreement with this finding, Weivoda et al. found that T2DM subjects treated with Dmab depicted both a significant reduction of dipeptidyl peptidase-4 and an elevation of glucagon-like peptide 1, leading to a greater improvement of HbA1c than subjects treated with Bps or calcium and vitamin D supplementation [60]. However, long-term beneficial effects have not been reported [59].

Despite the above-mentioned positive association between sclerostin levels and glucose metabolism, the sclerostin inhibitor **romosozumab** has not been reported to adversely affect glucose homeostasis to date.


***Summary:***
*Contrary to indications deriving from preclinical models, bone active therapies and their modulation of bone turnover do not appear to play a clinically significant role in glucose metabolism in humans (*
Table 1
*).*


## 4. Antifracture Efficacy of Antiosteoporotic Agents in Patients with T2DM

Several studies in diabetic animals have provided indirect evidence of a reduced risk of fracture with antiosteoporotic treatment through positive effects on their bone density, mass, and strength. In a rodent model of T2DM, risedronate treatment decreased the osteoclast number and impaired osteoclast function, and increased the vertebral bone mineral content (BMC) and femoral diaphysis BMD, as well as the mechanical strength in the vertebrae [61]. In the same animals, TPTD treatment increased the osteoblast number and function in vertebral trabecular bone, augmented the trabecular bone mass, and improved the vertebral BMD and mechanical strength. Furthermore, TPTD improved the cortical bone structure and increased the cortical BMD [61]. In streptozotocin-induced diabetic rats, zoledronate, alendronate, and raloxifene have shown antiresorptive effects, namely, a decrease in the bone turnover rate, an increase in BMD, and improved bone mechanical strength [62,63]. Finally, in Zucker diabetic fatty (ZDF) rats, the sclerostin inhibitor romosozumab has been shown to increase bone mass and strength and improve bone defect regeneration [64].

In humans, information regarding the efficacy of antiosteoporotic agents in T2DM is mainly provided by post hoc analyses of diabetic subgroups from large osteoporosis randomized clinical trials (RCTs) and some observational studies (Table 2).

Although diabetes is characterized by a low-turnover state, further reduction of bone turnover with antiresorptives does not seem to negatively affect the fracture-preventive potential [72]. In a retrospective study of a large Danish cohort, a similar antifracture efficacy level was found between diabetic and nondiabetic patients, as well as between T1DM and T2DM patients treated with antiresorptives, mostly ALN or raloxifene [72]. Similarly, in an analysis of data from the Danish national prescription registry, having diabetes, with or without complications, did not influence the risk of fracture in patients adherent to ALN treatment [75]. In a post hoc analysis of the diabetic patients included in the Fracture Prevention Trial (FIT), three years of ALN treatment increased BMD at all sites compared to a placebo group, and the increase did not differ from the respective increase in nondiabetic participants [75]. In addition, increased BMD at the lumbar spine (LS) with no change at the total hip (TH) after 12 months of bisphosphonate (mostly ALN) treatment was reported in a small, retrospective study in both patients with T2DM and nondiabetic controls [74]. In contrast, in another small retrospective study, BMD at the LS increased similarly in T2DM and nondiabetic patients receiving ALN treatment, but BMD at the TH, femoral neck (FN), and forearm decreased in the diabetic group, while it increased in the nondiabetic patients [73]. In a post hoc analysis of three RCTs, one year of risedronate treatment was shown to have similar effects on BMD and bone markers between diabetic and nondiabetic Japanese women [66]. Data regarding the efficacy of intravenous bisphosphonates (ibandronate, pamidronate, and zoledronate) in diabetic women are lacking. Of note, these agents are contraindicated in diabetic patients with significant renal impairment.

Raloxifene has shown similar [68,72] or even higher [67] efficacy in reducing the vertebral fracture risk in diabetic compared with nondiabetic women. The nonvertebral fracture risk has not been found to be affected by raloxifene treatment in diabetic patients, as is the case in nondiabetic women.

In a post hoc analysis of the FREEDOM and FREEDOM Extension studies on osteoporotic women with diabetes, Dmab was found to significantly increase BMD and decrease the vertebral but not the nonvertebral fracture risk compared with a placebo group [69]. In particular, in both FREEDOM and its extension, BMD at all skeletal sites (LS, TH, and FN) increased significantly compared to the placebo group, irrespective of the presence of diabetes. With regard to fractures, at the end of FREEDOM (three years), Dmab treatment had significantly reduced the risk for new vertebral fractures, but, surprisingly and in contrast with the results from the overall study population, a higher incidence of nonvertebral fractures was found in patients with diabetes treated with Dmab compared to those in the placebo group, which was mostly attributed to fractures of the forearm and ribs during the second year of the study [69]. The number of hip fractures was small, and no significant difference was found with denosumab treatment. The rate of nonvertebral fractures in the placebo group was lower in the diabetic compared with the nondiabetic subjects in FREEDOM. In contrast, during the extension, the incidence of both new vertebral and nonvertebral fractures remained low with long-term denosumab treatment in patients with diabetes, while nonvertebral fracture incidence returned to levels comparable to the placebo group during the subsequent seven years of follow-up in the same population [69]. Additionally, Dmab may improve muscle mass and strength [20] and thus may ameliorate sarcopenia in T2DM patients, thereby contributing to a reduction in falls, which could result in a reduction in the risk of fracture.

Since diabetes is characterized by low bone turnover, anabolic agents that stimulate bone formation may have an advantage over antiresorptives, at least from a pathophysiological point of view. In diabetic patients, TPTD treatment has been shown to achieve similar increases in LS and TH BMD and higher increases in FN BMD, along with a comparable nonvertebral fracture incidence compared with nondiabetic patients [71]. Real-world data have shown that TPTD is more effective in reducing clinical fracture in patients with diabetes (−77%) than in nondiabetic patients (−48%) [76]. In diabetic women, abaloparatide, a parathyroid hormone-related protein (PTHrP) analog available in the U.S. but not in the EU, has been shown to significantly increase BMD at the LS, FN, and TH, as well as the trabecular bone score (TBS), and result in a reduced fracture incidence compared to a placebo group [70]. The fracture incidence and changes in BMD have been shown to be comparable between diabetic patients treated with abaloparatide and TPTD [77]. As mentioned above, romosozumab improved the bone status of diabetic rats [64]; however, the increase in cardiovascular events compared to ALN reported in a phase 3 RCT raised safety concerns, especially in diabetic populations [77].

Of note, poorly managed or yet undetected diabetes is associated with a higher risk of drug-related osteonecrosis of the jaw, a well-known rare complication of long-term antiresorptive treatment [78].

In conclusion, bisphosphonates represent the first-line drugs for osteoporosis treatment in T2DM patients. Denosumab may be preferred in older patients and those with polypharmacy and/or impaired renal function. Given the low bone turnover and poor bone material properties of diabetic bone, osteoanabolic drugs could be an option in selected cases. Bone turnover markers do not predict fracture risk in diabetic subjects, therefore their use in the clinical follow-up of diabetic patients on antiosteoporotic medications may not be advisable [79].


***Summary:***
*Post hoc analyses of diabetic subgroups of large osteoporosis RCTs seem to suggest that antiosteoporotic drugs are effective for the management of bone health impairment in patients with diabetes.*


## 5. Conclusions

The osteokines produced during the bone remodeling process may influence glucose metabolism. Uncarboxylated osteocalcin improves both glucose tolerance and energy expenditure, possibly contributing to the prevention of T2DM. In mice, RANKL seems to worsen insulin sensitivity, while the decoy receptor OPG, which physiologically antagonizes the activity of RANKL on osteoclast precursors, is able to delay the onset of hyperglycemia in diabetic animals. RANKL inhibition with Dmab has not clearly been proven beneficial for glucose homeostasis in humans.

Based on experimental evidence, Wnt signaling plays an important role in the modulation of glucose metabolism. Indeed, the endogenous inhibitors of Wnt/β-catenin signaling pathway, sclerostin and DKK-1, are involved in adipogenesis, energy expenditure, and insulin sensitivity. In particular, sclerostin is associated with lower levels of insulin, alongside reduced insulin resistance in diabetic patients compared to controls. However, sclerostin inhibition in humans has not been reported to affect glucose homeostasis.

Bone morphogenetic proteins participate in the control of adult glucose homeostasis and energy expenditure and may prevent the development of diabetic complications in adult T2DM mice. A study in humans reported lower levels of circulating BMP-9, associated with both higher HbA1c and fasting blood glucose, in patients with T2DM compared to healthy controls.

At the beginning of this narrative review, we set out to address two scientific queries: (1) Do antiosteoporotic drugs exert an effect on glucose metabolism, and (2) do they have an impact on fracture risk reduction in diabetic patients with osteoporosis?

It has been amply demonstrated that antiosteoporotic drugs profoundly change bone turnover, but regarding the first question, the few available clinical studies on their effects on glucose homeostasis do not allow drawing definite conclusions in humans. Studies on bone anabolic agents have yielded discrepant results. While intermittent PTH (1–84) was reported to improve glycemic control, teriparatide appeared to increase both insulin resistance and the fasting glucose concentration. Some small studies with Bps in diabetic and prediabetic patients have shown improvements in glycemic parameters, together with a decreased risk of diabetes onset, but large registrative RTCs on ZOL and ALN have failed to confirm these observations. Although a recent paper has demonstrated that Dmab may induce a short-term positive effect on insulin resistance in postmenopausal women and women with breast cancer treated with an aromatase inhibitor, data from post hoc analyses on the FREEDOM trial did not confirm a positive action of Dmab on glycemic outcomes.

With regard to the second question, several studies on diabetic animals have provided solid evidence of a reduction in fracture risk with antiosteoporotic treatments, based on their ability to increase bone mineral density and, ultimately, to enhance bone strength. In humans, the available data are scarce and have been obtained from small observational studies or from post hoc analyses of large osteoporosis RCTs. Based on available data, Bps may be considered as the first-line osteoporosis treatment in DM patients, while Dmab is preferred for older patients, particularly those with impaired renal function. The low bone turnover state and the poor quality of diabetic bone suggest that osteoanabolic agents could be used, especially in patients with more severe forms of osteoporosis.

In conclusion, not enough data have been procured in order to draw definite conclusions with regard to the effects of antiosteoporotic drugs on glucose metabolism or their antifracture potential in the setting of diabetes mellitus.

Following da Vinci’s suggestion, more “experience” is needed to reach “wisdom” in this field.

## Figures and Tables

**Figure 1 jcm-10-00996-f001:**
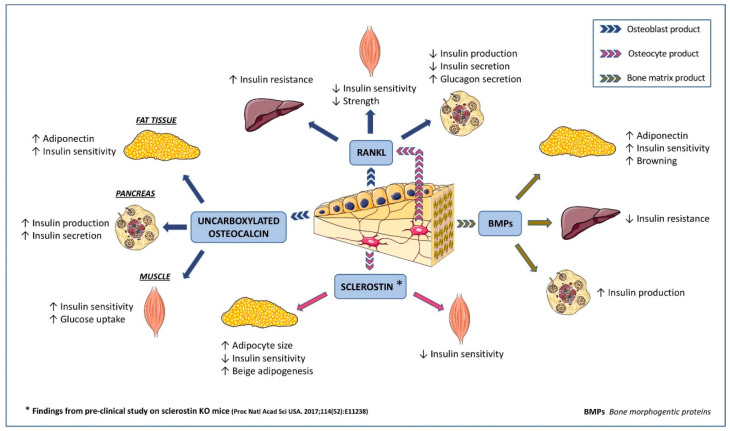
Potential action of main osteokines on glucose homeostasis *(see text for References)*.

**Table 1 jcm-10-00996-t001:** Clinical studies focused on the effect of antiosteoporotic agents on glucose metabolism.

Type of Study	Aim	Study Population	Main Findings	Main Limitations	Reference
		(Patients/Controls)			
Prospective, open-label study	The acute and chronic effect of TPTD on blood glucose and insulin	23 postmenopausal women with osteoporosis	TPTD seemed to have an acute, subclinical adverse impact on stimulated glucose levels. This impact tended to subside when TPTD was continued on a chronic basis.	Small sample size	Anastasilakis, 2007 [42]
Prospective, open-label study	Effects of intermittent TPTD versus the chronic exposure to excess endogenous PTH, as in PHPT, on glucose homeostasis	44 * postmenopausal osteoporotic nondiabetic women (25TPTD/22)	TPTD did not affect glucose homeostasis while in pHPT the continuously elevated Ca and endogenous PTH levels affected insulin sensitivity and resulted in increased insulin secretion.	Small sample size, the assumption that the two molecules are comparable regarding their effect on glucose homeostasis	Anastasilakis, 2008 [43]
Prospective, open-label study	The effect of TPTD on glucose metabolism	23 postmenopausal women with osteoporosis	Teriparatide may adversely affect some parameters of glucose metabolism, inflammation, and endothelial function.	Small sample size, lack of placebo group	Celer, 2016 [44]
Prospective, open-label study	To investigate whether treatment of GIO with Bps or TPTD may influence glucose metabolic control	111 subjects with GIO (45 Bps/33 TPTD/22 controls)	Teriparatide was shown to be associated with improvement in serum HbA1c.	No measurements of insulin secretion and sensitivity; no data on incidence of T2DM	Mazziotti, 2014 [45]
Parallel, randomized controlled, open-label trial	Effect on glucose homeostasis, fat distribution, and adipokine production during PTH (1–84)	46* postmenopausal osteoporotic nondiabetic women (24PTH/22 CaVit D)	PTH (1–84) increased osteocalcin and decreased blood glucose, without influence on insulin secretion, resistance, and pancreatic β cell function.	Small sample size, lack of placebo group and direct measure of insulin resistance	D’Amelio, 2015 [49]
Retrospective cohort study	The influence of ALN usage on the incidence of T2DM among osteoporotic patients	Osteoporotic subjects without DM (1011 ALN/3033)	The nonexposed group had a significantly higher incidence of DM.	Lack of information on significant risk factors for DM	Ding-Cheng Chan, 2015 [51]
Retrospective cohort study	The effect of exposure to Bps on the risk of incident T2DM	Osteoporotic subjects without T2DM (35,998 Bps/126,459)	The risk of incident T2DM was significantly lower in patients exposed to Bps compared to matched controls.	Lack of information on significant risk factors for DM including concomitant medications	Toulis KA, 2015 [50]
Post hoc of 3 RCTs	To test whether antiresorptive therapies result in higher FSG, increased weight, or greater DM incidence	FIT # (3084 ALN/3067); HORIZON-PFR ## (3537 ZOL/3576), FREEDOM ### (3535Dmab/3541)	Antiresorptive therapy did not have a clinically important effect on FSG, weight, or DM risk in postmenopausal women.	Design of the studies	Schwartz, 2013 [52]
RCT	Effect of ALN on plasma glucose, insulin indices of postmenopausal women with prediabetes and osteopenia	60 postmenopausal women (30 ALN/30)	ALN improved fasting plasma glucose, HbA1c, and insulin indices.	Small sample size	Fard, 2019 [54]
Prospective, open-label study	Effect of ALN on daily needs of insulin in patients with senile T1DM and osteoporosis	Women with T2DM and PMO (20 ALN/20)	ALN reduced the daily dose of insulin.	Small sample size	Maugeri, 2002 [55]
Prospective, open-label study	Effects of decreasing osteocalcin through RIS treatment on glucose homeostasis	84 postmenopausal women without DM	Risedronate reduced osteocalcin but this change was not associated with changes in glucose metabolism.	Small sample size, OGTT was not performed to test glucose homeostasis	Hong, 2013 [53]
Prospective, open-label study	Effects of Dmab on glucose metabolism in women with severe osteoporosis	14 postmenopausal c nondiabetic women with severe osteoporosis	A single 60 mg dose of denosumab did not induce clinically evident glucose metabolic disruption.	Small sample size, lack of a control group; single treatment	Passeri, 2015 [58]
Prospective, open-label study	Effects of Dmab on glucometabolic parameters, insulin resistance, and lipid profile in nondiabetic women	48 postmenopausal nondiabetic women	Dmab was not associated with relevant modification of insulin resistance and lipid profile.	Small sample size, lack of a control group, single treatment	Lasco, 2016 [57]
Post hoc of RCT	Effect of Dmab compared to placebo on FSG in women with DM or prediabetes enrolled in FREEDOM trial	*Diabetes* (342 Dmab/324) *Prediabetes* (628 Dmab/640)	Dmab did not affect FSG in PMO women with prediabetes or DM. Modest FSG lowering with Dmab in those with DM who were not on ADM.	Design of the study; HbA1c not available; no detailed information on ADM	Napoli, 2018 [29]
Prospective, open-label study	Effect of a single administration of 60 mg Dmab on glucose metabolism in a cohort of women with breast cancer treated with aromatase inhibitors	15 Postmenopausal nondiabetic women	Although Dmab induced a short-term positive effect on insulin sensitivity, a benefit on long-term glucose metabolism was not evident.	Small sample size, short-term investigation after only one dose of Dmab, lack of a control group	Rossini, 2020 [59]

* 24 subjects treated with PTH 1–84 vs. 22 subjects treated with calcium and vitamin D. ^#^ Fracture Intervention Trial; ^##^ Health Outcomes and Reduced Incidence with Zoledronic Acid Once Yearly Pivotal Fracture Trial; ^###^ Abbreviations: HbA1c, glycated hemoglobin; PTH, parathyroid hormone; PHPT, primary hyperparathyroidism; GIO, glucose-induced osteoporosis; DM, diabetes mellitus; T2DM, Type 2 diabetes mellitus; PMO, postmenopausal osteoporosis; ALN, alendronate; RIS, risedronate; Bps, bisphosphonate; Dmab, denosumab; TPTD, teriparatide; FSG, fasting serum glucose. ADM, antidiabetic medication; CaVit D, calcium and vitamin D supplements; OGTT, oral glucose tolerance test.

**Table 2 jcm-10-00996-t002:** Clinical studies focused on the effect of antiosteoporotic agents on fracture risk in diabetic patients.

Type of Study	Aim	Study Population	Main Findings		Main Limitations	Reference
		(Patients/Controls)	BMD	Fx		
Post hoc RCT	BMD changes after 3 years of ALN vs. placebo	women with T2DM and hip T-score ≤ −1.6 (148/149)	↑ vs. placebo at LS 5.7%; at TH 4.3%; at FN 3.4% (vs. ↑ 6.2% at LS; 4.3% at TH; 3.8% at FN in non-DM, *p* = NS)	NR	post hoc analysis, no Fx data, self-reported T2DM in a number of patients	Keegan, 2004 [65]
Post hoc RCT	LS BMD changes after 1 year RIS	men and women with T2DM and osteoporosis (53/832)	↑5.4% similar response with non-DM	NR	data from 3 different phase III RCTs, no Fx data, no data on hip BMD, small number of patients, only Japanese	Inoue, 2016 [66]
Post hoc RCT	VFx after 3 years of RLX vs. placebo in DM	women with DM and PMO (124/45)	NR	↑ efficacy on VFx reduction in DM compared to non-DM	post hoc analysis, no DXA data, no differentiation between T1DM/T2DM	Johnell, 2004 [67]
Post hoc RCT	VFx after 5.6 years of RLX vs. placebo in DM	women with DM not selected for low BMD (2300/2311)	NR	↓ clinical VFx in DM compared to placebo (1.3 vs. 1.9%, HR 0.65) No difference in non-VFx (8.5 vs. 8.7%) No difference between DM and non-DM patients	post hoc analysis, no DXA data, self-reported DM in a number of patients, no differentiation between T1DM/T2DM, inclusion of clinical (and not morphometric) VFx	Ensrud, 2008 [68]
Post hoc RCT	BMD, VFx, and non-VFx after 3 years of Dmab vs. placebo in DM vs. non-DM	women with DM and PMO (266/242)	compared to placebo: ↑ at LS 6.7%; at TH 6.1%; at FN 5.0% (vs. ↑ 9.1% at LS; 6.8% at TH; 5.6% at FN in non-DM, *p* = NS)	compared to placebo: ↓ new VFx (1.6 vs. 8%, HR 0.2); ↑ non-VFx (11.7 vs. 5.9%, HR 1.94)	post hoc analysis, no differentiation between T1DM/T2DM	Ferrari, 2020 [69]
Post hoc RCT	ABL vs.TPTD vs.placebo at 18 months in T2DM	women with DM2 and PMO (65//68*/65) *TPTD	ABL vs. placebo: ↑ at LS 7.6%; at TH 3.3%; at FN 2.8% *TPTD* vs. *placebo*: ↑ at LS 8.4%; at TH 2.7%; at FN 2.2% *ABL* vs. *TPTD*: NS *TBS ABL* vs. *TPTD* vs. *placebo*: 3.7% vs. 2.4% vs. −0.6	*ABL* vs. *TPTD* vs. *placebo*: Kaplan–Meier estimated rate for clinical Fx 3.6 vs. 5 vs. 7.4%, *p* = NS	post hoc analysis, Fx reported as AEs, small number of patients with T2DM	Dhaliwal, 2020 [70]
Post hoc prospective observational	efficacy of TPTD for 24 months on BMD and non-VFx risk in T2DM vs. non-DM patients	men and women with DM2 and osteoporosis (291/3751)	similar ↑ at LS and TH, and larger ↑ at FN in the DM2 compared to non-DM patients	similar non-VFx rate between DM2 and non-DM patients	post hoc analysis, a number of VFx were self-reported	Schwartz, 2016 [71]
Retrospective case-control	Effect of antiresorptives (BPs, mostly ALN and RLX) in DM	men and women with T1DM and T2DM (16,524/397,721)	NR	similar antifracture efficacy between DM and non-DM, and between T1DM and T2DM	retrospective, various antiresorptives	Vestergaard, 2011 [72]
Retrospective case-control	BMD after ALN for 4.8 years in T2DM vs. non-DM patients	women with T2DM and PMO (26/26)	compared to non-DM: similar ↑ at LS (5.5 vs. 4.8%); ↓ at TH (−5.6 vs. 1.4%) and FN (−8.1 vs. 1.1%) and forearm (−3.6 vs. 12.7%)	NR	retrospective, small number of patients, no Fx data	Dagdelen, 2007 [73]
Retrospective case-control	BMD changes 12 months after BP (mostly ALN) therapy in T2DM vs. non-DM patients	women with DM2 and PMO (35/35)	↑ at LS; unchanged at TH (no difference with non-DM patients)	NR	retrospective, small number of patients, no Fx data	Nan, 2016 [74]
Registry	major osteoporotic Fx risk after ALN therapy > 6 months	men and women treated with ALN	NR	DM did not influence Fx risk	observational, no BMD data, no differentiation between T1DM/T2DM	Abrahamsen, 2013 [75]

Abbreviations: ABL, abaloparatide; AE, adverse event; ALN, alendronate; BMD, bone mineral density; BP, bisphosphonate; Dmab, denosumab; DM, diabetes mellitus; FN, femoral neck; Fx, fracture; LS, lumbar spine; NR, not reported; NS, nonsignificant; non-VFx, nonvertebral fracture; PMO, postmenopausal osteoporosis; RIS, risedronate; RLX, raloxifene; T1DM, type 1 diabetes mellitus; T2DM, type 2 diabetes mellitus; TH, total hip; TPTD, teriparatide; VFx, vertebral fracture; vs., versus; ZOL, zoledronate.

## Data Availability

Data sharing not applicable.

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
