# Peer review of "The Impact of Antiosteoporotic Drugs on Glucose Metabolism and Fracture Risk in Diabetes: Good or Bad News?"

_jcm, 2021, doi:10.3390/jcm10050996_

Round 1

Reviewer 1 Report

I applaud the authors' work in synthesising the existing evidence-base on this topic. It is very comprehensive and the depth may not be easy to follow for the general generalist. Other comments:

  • Could the authors explain how the limitation in Table 1 came about? Was this what was documented in the respective papers or the authors' assessment of the paper?
  • Could A1c be converted to HbA1c? Many countries use the full term
  • Would the authors consider expanding the findings in Table 1 similar to what was done in Table 2 where effect size, etc is reported?
  • An explanation of how Type 1 and Type 2 DM is different in the context of osteoporosis and osteoporosis treatment would be helpful
  • Does glycaemic control impact on bone quality and effectiveness of osteoporosis treatment? The manuscript mostly looks at things the other way round.

Reviewer 2 Report

This narrative review investigates  anti-osteoporotic agents and glucose metabolism and fracture prevention in diabetes.

I have some comments:

1.Although it is a narrative review the search strategy would be improved by a date of the search. Also a systematic literature review is always stronger than a narrative review and the authors may consider to convert the review to a systematic review.

2.A limitation on cross-sectional studies investigating osteocalcin and glycemic markers is that many of these studies include both a non-diabetic and a diabetic group and investigate associations across the entire group. In type 2 diabetes the bone turnover marker levels are decreased compared to controls. Thus associations between osteocalcin and glycemic markers are likely to be driven by diabetes or no diabetes. This should be discussed in the present review.

  1. Were individuals matched on BMI in the Toulis 2015 study? Lack of matching on BMI may bias the results.

4.In diabetes the fracture risk is increased although BMD paradoxically does not explain this. A major drawback to most of the studies in table 2 is that patients with osteoporosis are investigated.  Thus, in patients with diabetes and osteoporosis the effect of anti-osteoporotic treatment is comparable to non-diabetic individuals without osteoporosis. What is the evidence on diabetes patients with normal/slightly decreased BMD and fracture prevention of anti-osteoporotic treatment?
